# Research on multiple bubbles in China's multi-level stock market

Ge Li[1], Ming Xiao [1]*, Xionghui Yang[1,2], Ying Guo[1], Shengyi Yang[1]

**1** School of Economics and Management, University of Science and Technology Beijing, Beijing, People's Republic of China, **2** CITIC Group Corporation, Beijing, People's Republic of China

These authors contributed equally to this work.
‡ These authors also contributed equally to this work.
\* blue_ridge111@yahoo.com

**Data Availability Statement:** All relevant data are within the paper and its S1–S4 Files.

**Funding:** The author(s) received no specific funding for this work.

## Abstract

Financial bubbles have always been a topic of long-term concern for economists. Understanding bubble phenomenon and dating the period of bubbles in real time can provide an early warning diagnosis for financial bubbles and help regulatory authorities to control it and maintain market order. The generalized sup ADF (GSADF) and backward sup ADF (BSADF) tests with flexible window width can effectively detect and date periodically collapsing bubbles in real time. Based on the financial present value model, this paper applies right-tail recursive ADF test to test multiple bubbles in China's multi-level stock market. Unlike the other researches in China, the ratios of the real stock prices' natural logarithm to the real dividends' natural logarithm are used for our testing instead of stock price index. Empirical results show that there are 8 bubbles in the Main-Board Market, 6 bubbles in the Small and Medium Enterprises Board (SMEs), and 4 bubbles in the Growth Enterprise Market (GEM). These bubbles are liquidity-driven and presuppose a loose credit cycle, with the exception of bubbles in 2014–2015. The frequent emergence of bubbles in a short time indicates that China's stock market is still emerging market. In addition, frequent fluctuations imply there is a serious "herd effect" and a lack of monitoring mechanism for bubble risk. This study not only enrich the real-time dynamic research on periodical bubbles of China's stock market, but also provide an empirical reference for investors' investment choices, financial decisions of listed companies and warning mechanism of regulatory authorities.

## 1. Introduction

A large number of studies have shown that asset price bubbles and their subsequent collapse can lead to misallocation of resources and may have a serious impact on the country's actual economic activities. The 2008–2009 global financial turmoil triggered by US subprime mortgage crisis and its subsequent impact on commodity markets, exchange rates and real economic activity has rekindled economists' interest in researching financial bubbles and its potential global consequences. Empirical tests need to provide early warning diagnosis for

**Competing interests:** The authors have declared that no competing interests exist.

bubbles to help central banks, regulators and policy makers maintain economic and financial stability.

Skyrocketing price of Dutch tulip bulbs in the 1630s is regarded as the first bubble recorded in economic history. The Internet stock bubble at the turn of the century [1], the real estate bubble [2], and the Bitcoin bubble [3] are all the recognized bubble in the world. Asset price bubble can be divided into two categories: rational bubble and irrational bubble. Irrational bubble have been studied from the perspective of behavioral finance and game theory by combining investor psychology with environmental factors [4, 5], including fashion trend model, positive feedback investment strategy model, noise trader model, investor sentiment model and herd behavior model.

Rational bubble theory assumes that the actual price of an asset is equal to the sum of the present value of relevant fundamentals and the bubble component which is expected to grow at the real interest rate, meanwhile the investors have rational expectations. Early rational bubble test based on the assumption that the bubble was linear, which meant that the bubble would always exist and would not burst or start again. Variance bound tests [6, 7] imposed an upper bound on stock price volatility. Once actual price volatility exceeded the bound, there might be a bubble. In fact, this method was a test of the validity of standard present value model, the rejection of null hypothesis may be due to any assumption of the model failure. Marsh and Merton [8] suggested that variance bound tests would fail when dividends and stock prices were non-stationary, and along with the small sample distortions. West's two-step tests [9, 10] were a milestone in asset price bubble test, the existence of bubbles was explicitly considered as alternative hypotheses for the first time. This method tested for the presence of a bubble on the basis of comparing asset price model with and without bubbles. Dezbakhsh and Demirguc-Kunt [11] criticized West's approach for size distortion in small samples, rejecting null hypotheses frequently, and being inconsistent under bubble alternative hypothesis. In addition, if there is a relatively short-term bubble in a long time series of data, the existence of this bubble may not be correctly recognized.

However, there are certain theoretical properties that can be clearly used for bubbles detection. Traditional unit roots and cointegration tests have also been widely used for bubble testing. If first differences of stock prices have a stationary mean and/or stock prices are cointegrated with dividends, there would be evidence against the existence of rational bubbles. Diba and Grossman [12, 13] observed that rational bubble could not start (or restart); therefore, if there is a bubble, it must exist on the first day of trading and always exist. If the bubble "pops", the bubble must collapse to zero due to the lack of arbitrage opportunities and the impossibility of negative prices. They applied the Dickey-Fuller (DF) test to the price and dividend series in levels and first differences, then found that both stock prices and dividends were stationary in first differences. Since differencing an explosive autoregressive process does not lead to a stationary process, a rejection from the DF test for the first difference of the price and dividend series, suggests that no rational bubble exists. When they conducted a cointegration test [14], they still found stock prices and dividends were cointegrated, indicating no rational bubble exists. Evans [15] disagreed with Diba and Grossman's argument, he believed that the bubble may break into a small non-zero value, and then continue to grow. For the first time, he proposed a bubble model of periodically collapsing. He pointed out that it was difficult to detect collapsing bubbles based on traditional unit root tests because they behaved more like a stationary process than an explosive process.

Since then, many papers attempted to overcome the difficulty of detecting periodic collapsing bubbles and the nonlinear method started to be applied to bubble testing. The Monte Carlo experiment conducted by Hall et al. [16] shows that the two-regime Markov switching ADF test with constant switching probabilities performs well in bubble testing. However, the

Markov model is computationally expensive, and the asymptotic distribution of unit root test statistic calculated by this model is hard to analyze in some cases.

In recent years, the breakthrough of the periodical bubbles test is that the right-tailed DF test is applied to relevant series. The null hypothesis is unit root process, and the alternative hypothesis is explosive autoregression process. Phillips, Wu and Yu (hereafter PWY) [17] proposed a right recursive unit root test, Sup ADF (SADF), which can detect the exuberance of asset price series during the bubble expanding period. Additionally, PWY time-stamping strategy can identify points of origination and termination of a bubble. But the estimation is consistent when there is only one bubble in data. When the sample period is long enough, there will be multiple bubbles. The complex nonlinear structure break formed from multiple collapsing bubbles could diminish the discriminatory power of SADF test. Therefore, Phillips, Shi and Yu (hereafter PSY) [18] provided a new framework for testing and dating multiple bubbles, namely Generalized Sup ADF with flexible window width. PSY then proposed a new time-stamping strategy based on a backward regression technique, namely Backward Sup ADF, to identify the origination and termination points of multiple bubbles, and the estimated points are consistent with the actual points [19, 20]. The recursive evolving algorithm relies only on historical information and allows time-varying model structures, which has general applications in regression. Different from the regime switching, it is a real-time program and easy to implement. This real-time method has been recognized by central bank economists and financial regulators and applied to a variety of markets as an early exuberance warning system [21–24].

Besides, other methods such as the revised Bhargava statistic [25], the modified Busetti-Taylor statistic [26], and the modified Kim statistic [27] also have same recursive features as SADF test. Other non-linear procedures, such as Chow test and CUSUM test can also be used as time stamping methods. But Homm and Breitung [28] have proved through a large number of simulations that PWY procedure has higher test satisfaction than other recursive programs when there are structural breaks.

There are several methods for testing China's asset price bubble. A simpler method is to judge whether there is a bubble by the value of a certain indicator, such as the P/E ratio and the Tobin Q ratio. But it cannot comprehensively reflect the real situation of bubbles. The fundamental value comparison is also used to test bubbles by comparing the fundamental value of asset with its actual price [29]. Nonlinear methods are also widely used to detect bubbles in China's stock market. For example, Deng et al. [30], Zhang, Xu and Zhai [31] and Yu and Ma [32] adopted SADF test to conduct bubble testing and dating on China's stock market. Most of the researchers only adopt price series to detect China's capital market bubble without considering dividends when they use the right-tail recursive unit root test proposed by PWY [31, 32]. It does not conform to the specification of test model and cannot be theoretically supported.

Based on above analysis, this paper uses the right-tail recursive unit root test with flexible window width to carry out real-time dynamic research of bubbles in China Shanghai and Shenzhen 300 (CSI 300) Index, SMEs Index and GEM Index respectively. Unlike the other researches in China, in this paper, ratios of the real stock prices' natural logarithm to the real dividends' natural logarithm are selected as target sequences, and critical values are obtained by 2000 times Monte Carlo simulation, thus we can date the start and end points of periodical bubbles in real time.

The rest of this paper is organized as follows. In Section 2, the basic model specification of bubble test and the principles of right-tail recursive ADF test are discussed, including SADF, GSADF and BSADF. In Section 3, the sample processing is introduced and the relevant preliminary data analysis is provided. The empirical bubble test results and result analysis are introduced in Section 4. In Section 5, the full text is summarized and conclusions are drawn.

## 2. Theoretical model

### 2.1 Model specification

The concept of rational bubble can be illustrated by financial present value theory, where the fundamental asset price is determined by the sum of current discounted values of expected future dividend series [33, 34]. Most tests begin with the following standard no-arbitrage conditions [35]:

$$P_t = \frac{1}{1+r_f} E_t(P_{t+1} + D_{t+1}) \tag{1}$$

Where $P_t$ represents the real stock price adjusted by dividends at time $t$. $D_t$ represents the real dividend obtained during the holding period from time $t$-1 to $t$. $f$ means free, $r_f$ stands for risk-free rate of return ($r_f > 0$), namely, investors expect a constant return on assets. $1/(1+r_f)$ is the discount rate.

PWY follow the research done by Campbell and Shiller [36], using the log-linear approximation of Eq (1) to obtain following solutions by recursive substitution:

$$p_t = p_t^f + b_t \tag{2}$$

$$p_t^f = \frac{\kappa - \gamma}{1-\rho} + (1-\rho) \sum_{i=0}^{\infty} \rho^i E_t d_{t+1+i}$$
$$b_t = \lim_{i \to \infty} \rho^i E_t p_{t+i} \tag{3}$$

$$E_t(b_{t+1}) = \frac{1}{\rho} b_t = \left(1 + \exp(\overline{d-p})\right) b_t \tag{4}$$

Where $p_t = \log(P_t)$, $d_t = \log(D_t)$, $\rho$ is the discount rate, with $\rho = 1/(1+r_f)$, and obviously, $0 < \rho < 1$. $\gamma = \log(1+r_f)\rho = 1/(1 + \exp(\overline{d-p}))$, with $\overline{d-p}$ is the average log dividend-price ratio. $\kappa = -\log(\rho)-(1-\rho)\log(1/\rho-1)$.

By convention, $p_t^f$ is called the fundamental component of stock price, which is determined by the expected dividend; $b_t$ is the rational bubble component, which satisfies the difference Eq (5) below. Both components are expressed in natural logarithms. Since $\exp(\overline{d-p}) > 0$, $b_t$ is a submartingale and expected to be explosive. Eq (4) means following process:

$$b_t = \frac{1}{\rho} b_{t-1} + \varepsilon_{b,t} \equiv (1+g)b_{t-1} + \varepsilon_{b,t}, \ E_{t-1}(\varepsilon_{b,t}) = 0 \tag{5}$$

Where $g = 1/\rho - 1 = \exp(\overline{d-p}) > 0$ is the growth rate of the natural logarithm of bubble, and $\varepsilon_{b,t}$ is a martingale difference.

It is apparent from Eq (2) that the stochastic properties of $p_t$ are determined by those of $p_t^f$ and $b_t$.

If there is no bubble, i.e. $b_t = 0$, $\forall t$, then $p_t = p_t^f$, the properties of $p_t$ are only determined by those of $p_t^f$. In this case, from Eq (3), we can get:

$$d_t - p_t = -\frac{\kappa - \gamma}{1-\rho} - \sum_{i=0}^{\infty} \rho^i E_t(\Delta d_{t+1+i}) \tag{6}$$

If both $p_t$ and $d_t$ are first-order intergrated processes, i.e. $I(1)$, then Eq (6) implies $p_t$ and $d_t$ are cointegrated and have a cointegration vector $[1,-1]$.

If there is a bubble, i.e. $b_t \neq 0$, it can be seen from Eq (2) that whether $d_t$ is a first-order intergrated process $I(1)$ or stationary process $I(0)$, $p_t$ will be explosive. In this case, $\Delta p_t$ is also explosive and not a stationary process.

It can be seen from Eqs (2) and (5) that a direct way to test bubbles is to test the explosive behavior of $p_t$ and $d_t$ under the situation that the discount rate is constant [37], since $p_t = \log(P_t)$, $d_t = \log(D_t)$, which is to test the explosive behavior of the natural logarithm of stock prices and the natural logarithm of dividends. If the explosive characteristics of $p_t$ are generated by $d_t$, then these two processes will be explosive cointegration. If $d_t$ is non-explosive, the explosive behavior in $p_t$ will provide sufficient evidence for the presence of bubble. PSY's tests of US market bubbles [18] inspired this paper to test bubble component by examining the explosiveness of the ratio of the natural logarithm of stock prices to the natural logarithm of dividends.

Explosiveness means that there is an explosive root in the autoregressive expression of time series, that is, in a sub-period of the sample, the coefficient $\delta$ in first-order autoregressive process $x_t = \alpha_x + \delta x_{t-1} + \varepsilon_{x,t}$ satisfies $\delta > 1$, where $\alpha_x$ is a constant term, $\delta$ is an autoregressive coefficient, and $\varepsilon_{x,t}$ is a residual term. In a certain sample, if $\delta > 1$, it conveys an explosive autoregressive behavior, which is bubble. The above AR (1) process is simulated by setting $\alpha_x = 0$, $\varepsilon_{x,t} \sim$ iid $N(0,1)$. Fig 1 shows the trajectories of simulated stationary time series ($\delta = 0.8$), random walk process ($\delta = 1$) and explosive autoregressive process ($\delta = 1.05$), thus, the principle of the test can be visually understood. It can be seen that the difference among these three trajectories is very obvious. When $\delta = 0.8$, it is a stationary process, and the time series fluctuates around the value of 0. When $\delta = 1$ (the null hypothesis of the test), it is a unit root process, i.e.

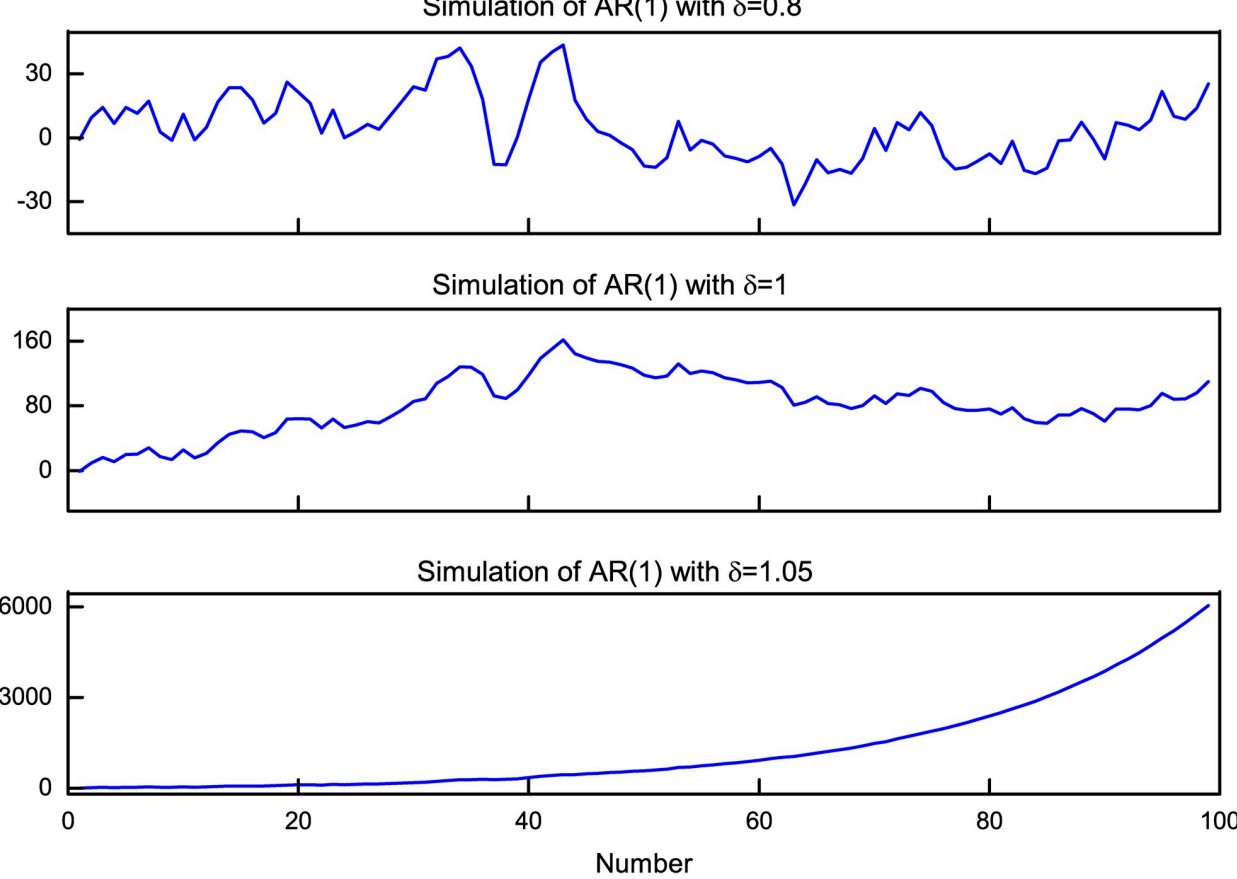

**Fig 1. Trajectories of typical stationary, random walk, and explosive autoregressive.**

a random walk process, which is used to express the irregular change form. In this case, every change is random, just like a random record of one's drunken walk. When $\delta = 1.05$ (the alternative hypothesis of the test), it is an explosive process, the time series increases rapidly over time.

## 2.2 The sup augmented Dickey-Fuller (SADF) test

As described by Evans [15], when the bubble collapses periodically, its collapsing behavior is transient, and its observed trajectory may seem more like an $I(1)$ process or even a stationary sequence, rather than an explosive sequence. Under this situation, the results of traditional ADF test will be confusing. But the SADF test proposed by PWY [17] can solve this problem. The SADF is a right tail unit root test, which focuses on the alternative hypothesis. Therefore, the null hypothesis and the alternative hypothesis are:

$H_0$: $\beta = 0$, time series is a unit root process (martingale process);

$H_1$: $\beta > 0$, time series is an explosive root process.

This method estimates autoregressive Eq (7) by the ordinary least squares. In applications of this paper, the optimal lag order $k$ is determined by BIC criterion. The initial lag order is set to 12, and finally the $k$ that minimizes the BIC value of the model is selected.

$$\Delta x_t = \alpha + \beta x_{t-1} + \sum_{i=1}^{k} \lambda_i \Delta x_{t-i} + \varepsilon_t \tag{7}$$

Where $\Delta$ is a difference symbol, $\alpha$ represents a constant term, $\beta$ is an autoregressive coefficient, $i$ denote the lag order, $k$ is an arbitrary positive integer, $\lambda_i$ is the coefficient of i-th lag term, and $\varepsilon_t$ is a residual term.

The recursive test performs a rolling window ADF test on total sample. The start point of rolling samples is the $r_1^{th}$ fraction of total sample $T$, and the end point is the $r_2^{th}$ fraction of $T$, where $r_2 = r_1 + r_w$, with $r_w$ (fraction) representing the recursive window width. Based on this, the recursive model can be rewritten as Eq (8):

$$\Delta x_t = \alpha_{r_1,r_2} + \beta_{r_1,r_2} x_{t-1} + \sum_{i=1}^{k} \lambda_{r_1,r_2}^i \Delta x_{t-i} + \varepsilon_t, \ \ \varepsilon_t \sim NID(0, \sigma_{r_1,r_2}^2) \tag{8}$$

The number of observations in each recursive subsample is $T_w = \lfloor Tr_w \rfloor$, and $\lfloor . \rfloor$ is the floor function. The ADF statistic ($t$-statistic) based on this subsample regression is denoted as $ADF_{r_1}^{r_2}$. The principle of SADF test is to repeat ADF test for each forward expanded sub-sample and take the maximum value of corresponding ADF test statistic sequence. In the recursion, $r_1$ is fixed at 0, $r_w$ changes from $r_0$ to 1 ($r_0$ is the initial window width, 1 is the full sample window width), and the end point $r_2$ of each subsample is $r_w$. The ADF statistic for a subsample from 0 to $r_2$ is $ADF_0^{r_2}$. Therefore, the SADF statistic is defined as (sup represents the maximum value):

$$SADF(r_0) = \sup_{r_2 \in [r_0, 1]} ADF_0^{r_2}$$

## 2.3 PWY time-stamping strategy

By comparing the SADF statistic with its corresponding right-tail critical value, the existence of bubbles can be tested. Furthermore, the origination and termination points of bubbles can be identified by comparing the recursive test statistic sequence $\{ADF_0^{r_2}\}_{r_2 \in [r_0, 1]}$ with their right

tail critical value sequence of the asymptotic distribution of $t$-statistics corresponding to standard ADF test.

Origination date $\lfloor Tr_e \rfloor$ of a bubble is calculated as: in the sequence of statistics $\{ADF_0^{r_2}\}_{r_2 \in [r_0, 1]}$, the first chronological observation point $r_e$ whose ADF statistic exceeds its critical value, and the calculated origination date is denoted by $\lfloor T\hat{r}_e \rfloor$. Termination date $\lfloor Tr_f \rfloor$ of a bubble is: after $\lfloor T\hat{r}_e \rfloor + \log(T)$, the first chronological observation point $r_f$ whose ADF statistic goes below its critical value, and the calculated termination date is denoted by $\lfloor T\hat{r}_f \rfloor$. PWY [17] indicate that the duration of a bubble must exceed a slowly varying amount of $L_T = \log(T)$, which helps to eliminate short-term fluctuations. The estimates of start and end points are constructed by following formulas:

$$\hat{r}_e = \inf_{r_2 \in [r_0, 1]} \left\{ r_2 : ADF_{r_2} > cv_{r_2}^{\beta_T} \right\} \tag{9}$$

$$\hat{r}_f = \inf_{r_2 \in [\hat{r}_e + \log(T)/T, 1]} \left\{ r_2 : ADF_{r_2} < cv_{r_2}^{\beta_T} \right\} \tag{10}$$

With $cv_{r_2}^{\beta_T}$ is the $100(1-\beta_T)\%$ critical value of ADF statistics based on $\lfloor Tr_2 \rfloor$ observations. The significance level $\beta_T$ is usually 0.05.

## 2.4 The generalized sup ADF test (GSADF)

The SADF test is very effective when there is only one bubble event in time series. When analyzing long-term data or samples containing multiple bubbles, the applicability of this method will be reduced. Therefore, PSY [18] proposed the GSADF test based on SADF. The GSADF test no longer fixes the starting point of subsamples to the first observation, but uses starting points of change. It allows recursive start and end points to vary within a possible range, making the window width more flexible. Because this test covers more subsamples within observations, the collapsing behavior of multiple bubbles can be detected more efficiently.

GSADF test still repeats ADF test on Eq (8), with the recursive end point $r_2$ from $r_0$ to 1, and the start point $r_1$ ranges from 0 to $r_2-r_0$. The maximum value of ADF test statistics in all feasible ranges from $r_1$ to $r_2$ is the GSADF statistic, which is denoted as:

$$GSADF(r_0) = \sup_{\substack{r_2 \in [r_0, 1] \\ r_1 \in [0, r_2 - r_0]}} \left\{ ADF_{r_1}^{r_2} \right\}$$

## 2.5 The backward sup ADF test (BSADF)

The BSADF is similar to GSADF and is a recursive test using flexible window widths. The difference between BSADF and GSADF is that BSADF performs a double recursive test on samples. BSADF test performs SADF test on each subsample that expands backward, where the end point of each subsample is fixed at $r_2$, the start point $r_1$ ranges from 0 to $r_2-r_0$, and the corresponding ADF statistic sequence is $\{ADF_{r_1}^{r_2}\}_{r_1 \in [0, r_2 - r_0]}$. The maximum value of ADF statistic sequence is defined as the BSADF statistic within this interval, i.e.:

$$BSADF_{r_2}(r_0) = \sup_{r_1 \in [0, r_2 - r_0]} \left\{ ADF_{r_1}^{r_2} \right\}$$

Fig 2 shows the comparison of sample sequence selections for the four tests (ADF, SADF, GSADF, and BSADF) to more intuitively understand the essential differences among them. Comparing GSADF (Fig 2c) with BSADF (Fig 2d), it can be found that GSADF statistic is the

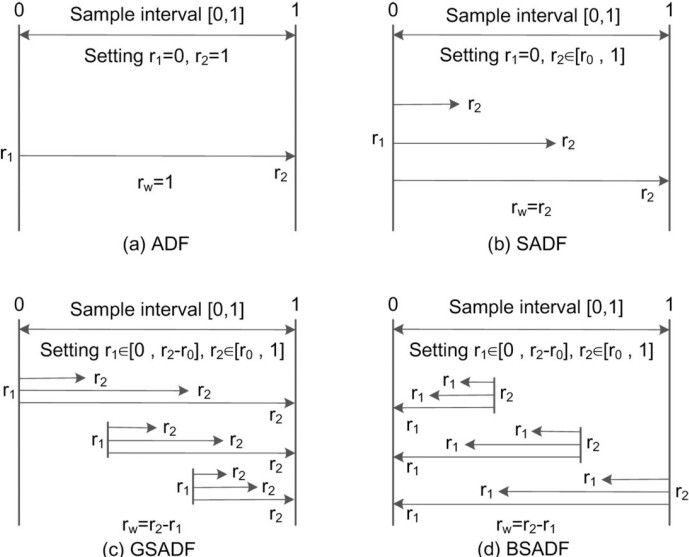

**Fig 2. Sample sequences of the four tests.** This figure compares the sample sequence selections for (a) ADF, (b) SADF, (c) GSADF, and (d) BSADF, where $r_0$, $r_1$, $r_2$, $r_w$ are the initial window width, start point, end point, and recursive window width of rolling samples, respectively.

maximum value of BSADF statistic sequence $\{BSADF_{r_2}(r_0)\}_{r_2 \in [r_0, 1]}$, namely:

$$GSADF(r_0) = \sup_{r_2 \in [r_0, 1]} \left\{ BSADF_{r_2}(r_0) \right\}$$

## 2.6 BSADF time-stamping strategy

Similar to PWY dating method, in BSADF statistic sequence $\{BSADF_{r_2}(r_0)\}_{r_2 \in [r_0, 1]}$, the first chronological observation point whose statistic exceeds its critical value is the start point of this bubble, after $\lfloor T\hat{r}_e \rfloor + \theta \lg(T)$, the first chronological observation point whose statistic goes below its critical value is the end point. It is assumed that the duration of a bubble is at least $\theta \lg(T)$, and $\theta$ is a frequency-dependent parameter. For example, in this paper, we restrict a bubble must last for more than 14 weeks, because $T$ of CSI 300, SMEs Board, and GEM are 735, 699, and 525 respectively, $\theta$ are 4.8844, 4.9218, and 5.1468 respectively. The start point $r_e$ and the end point $r_f$ are estimated by the first crossing time formulas (11) and (12):

$$\hat{r}_e = \inf_{r_2 \in [r_0, 1]} \left\{ r_2 : BSADF_{r_2}(r_0) > scv_{r_2}^{\beta_T} \right\} \tag{11}$$

$$\hat{r}_f = \inf_{r_2 \in [\hat{r}_e + \theta \lg(T)/T, 1]} \left\{ r_2 : BSADF_{r_2}(r_0) < scv_{r_2}^{\beta_T} \right\} \tag{12}$$

With $scv_{r_2}^{\beta_T}$ is the $100(1 - \beta_T)\%$ critical value of SADF statistics based on $\lfloor Tr_2 \rfloor$ observations. The significance level $\beta_T$ is usually 0.05.

## 3. Samples

### 3.1 P/E ratio

P/E ratio is an effective and important indicator for judging whether stocks are fairly priced. Under normal circumstances, P/E ratios of mature capital markets and emerging capital

markets are maintained at 10–20 and 20–30, respectively, while Wind statistics show that as of Aug 28, 2020, P/E ratios of CSI 300 Index, SMEs Index and GEM Index are 14.65, 37.82 and 68.26 respectively. The overall P/E ratio is high, which can be inferred there is a risk of stock overvaluation, indicating the possibility of bubbles.

## 3.2 Real stock prices and real dividends

Since dividends data of these three sectors cannot be directly obtained from available database, by referring to the method proposed by Cochrane [38], nominal dividend series of each sector can be deduced from the total return index ($R_t^T$) and the price-only index ($R_t^C$). Namely:

$$D_t = R_t^T - R_t^C \tag{13}$$

This paper selects the cross-market index, weekly closing prices of CSI 300 Index and CSI 300 Total Return Index, issued jointly by Shanghai Stock Exchange and Shenzhen Stock Exchange, as the source sample for Main-Board Market research, which can comprehensively reflect the overall trend of Shanghai and Shenzhen stock markets. The sample range is from April 7, 2006 to August 28, 2020, totaling 735 observations. For the small and medium enterprises board, weekly closing prices of SMEs Index and SMEs Total Return Index issued by Shenzhen Stock Exchange are selected as the source sample, which range from December 29, 2006 to August 28, 2020, with a total of 699 observations. Weekly closing prices of GEM Index and GEM Total Return Index issued by Shenzhen Stock Exchange are selected as source samples for GEM market. The sample range is from June 4, 2010 to August 28, 2020, with a total of 525 observations. All data are derived from Wind information.

The CSI 300 index, SMEs index and GEM index are the price-only index series, that is, the nominal stock price series. And the nominal dividend series can be calculated by Eq (13). This paper converts nominal series into real series through the Consumer Price Index deflator, which is derived from the National Bureau of Statistics of the People's Republic of China. The Consumer Price Index deflator of each board is a fixed base index based on the initial date of each sample. Fig 3 shows the changes in real stock price series and real dividend series of three sectors. It can be seen that three real price series are constantly fluctuating, while real dividend series are steadily rising in the fluctuation. However, rise in both price series and dividend series in the first half of 2015 seems unusual. In just half a year, real price indexes had more than doubled. This phenomenon implies that there may be periodically bubbles.

## 3.3 ln($p$)/ln($d$)

Most researchers just only use the prices as target sequences to test China's capital market bubbles, without considering dividends. If dividends are set aside, only investment is taken into

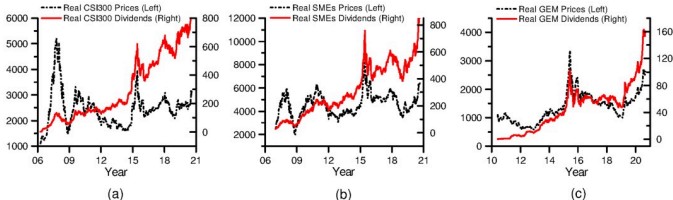

**Fig 3. Real price index and real dividend index.** This figure plots the real price index (left scale) and real dividend index (right scale) for three sectors: (a) CSI 300 from April 7, 2006 to August 28, 2020, (b) SMEs board from December 29, 2006 to August 28, 2020, (c) GEM from June 4, 2010 to August 28, 2020. All series data are weekly. Source: Wind database, and website of the National Bureau of Statistics of the People's Republic of China.

account, while income factors are neglected. If rise in stock price is accompanied by a rise in dividends, it means there is performance support in this company and cannot just judge bubbles from prices. For example, Company A's stock price is 100 Yuan per share, cash dividend is 10 Yuan per share, while company B's stock price is 10 Yuan per share, cash dividend is 1 cent per share. At this time, there are more likely bubbles in B's share price. If only considering the price 100>10, it is biased to say there is a bubble in company A's stock price.

Although China's stock market has a low dividend yield, dividend sequence has always existed. After more than 20 years of development, China's capital market has become more and more market-oriented and efficient. The dividends have been steadily increasing in the past decade. As can be seen from Fig 3, real dividend series all show a steadily rising trend in fluctuations.

Based on analysis of Section 2.1, the ratios of the real stock prices' natural logarithm to the real dividends' natural logarithm (i.e. $\ln(p)/\ln(d)$) are selected as the target series to analyze periodical bubbles of each sector in real-time. The calculated $\ln(p)/\ln(d)$ series are plotted as Fig 4. Due to the initial instability of stock market, dividends are less, or even no dividends, which results in large fluctuations of the initial $\ln(p)/\ln(d)$. With the improvement of market and the steady increase of dividends, the $\ln(p)/\ln(d)$ series tend to be relatively stable. In order to better observe the fluctuations of $\ln(p)/\ln(d)$ series, data of CSI 300 was intercepted from May 26, 2006 to August 28, 2020, and data of GEM was intercepted from April 15, 2011 to August 28, 2020. Entire samples of these two markets are shown in the small maps at the top right of each, while the entire sample for SMEs is directly showed in Fig 4.

It can be seen from Fig 4, remarkably different from the price–dividend ratio of S&P500-based series in PSY [18], series of three sectors all show a significant downward trend. This is because China's stock market is unstable at initial stage, the dividend distribution system is not perfect, and there is always tendency of "heavy financing, light return". Therefore, the overall dividend level is relatively low at the beginning, resulting in a larger value of $\ln(p)/\ln(d)$. However, following the promulgation of "Decision on Amending Certain Provisions on Cash Dividends of Listed Companies" in October 2008, China Securities Regulatory Commission issued a series of dividend policies to promote cash dividends and guide listed companies to improve their cash dividend mechanism. After the reform, dividends of most companies have increased significantly, so the $\ln(p)/\ln(d)$ series show a gradual decline trend.

## 4. Empirical test

### 4.1 Parameter setting

(1) **Initial window $r_0$.** In practice, the initial window width should be chosen according to the total observations $T$. If $T$ is small, then $r_0$ needs to be large enough to ensure there are enough observations to make a sufficient initial estimate; if $T$ is large, $r_0$ can be set smaller, so that the test will not miss any opportunity to detect early explosive behavior. In this paper, $r_0 = 0.01+1.8/T^{1/2}$, and the number of observations in initial window is $\lfloor Tr_0 \rfloor$. In calculation, the step size is one week. That is, after each recursion, a new observation is added into the window to form a new subsample for cyclic estimation. The $T$ of CSI 300, SMEs and GEM are 735, 699 and 525 respectively, so $r_0$ are 7.64%, 7.81% and 8.86% respectively, and the numbers of observations in initial windows are 56, 54 and 46 respectively.

(2) **Critical value.** Critical values of SADF test, GSADF test and time-stamping strategy in this study are all implemented by 2000 times Monte Carlo simulations. Wiener process approximates the partial sum of 2000 independent $N(0,1)$ variables, so when simulating critical value, firstly 2000 pseudo-random series of standard normal distribution which exactly match the number of test observations $T$ are generated, and the cumulative sum of these 2000

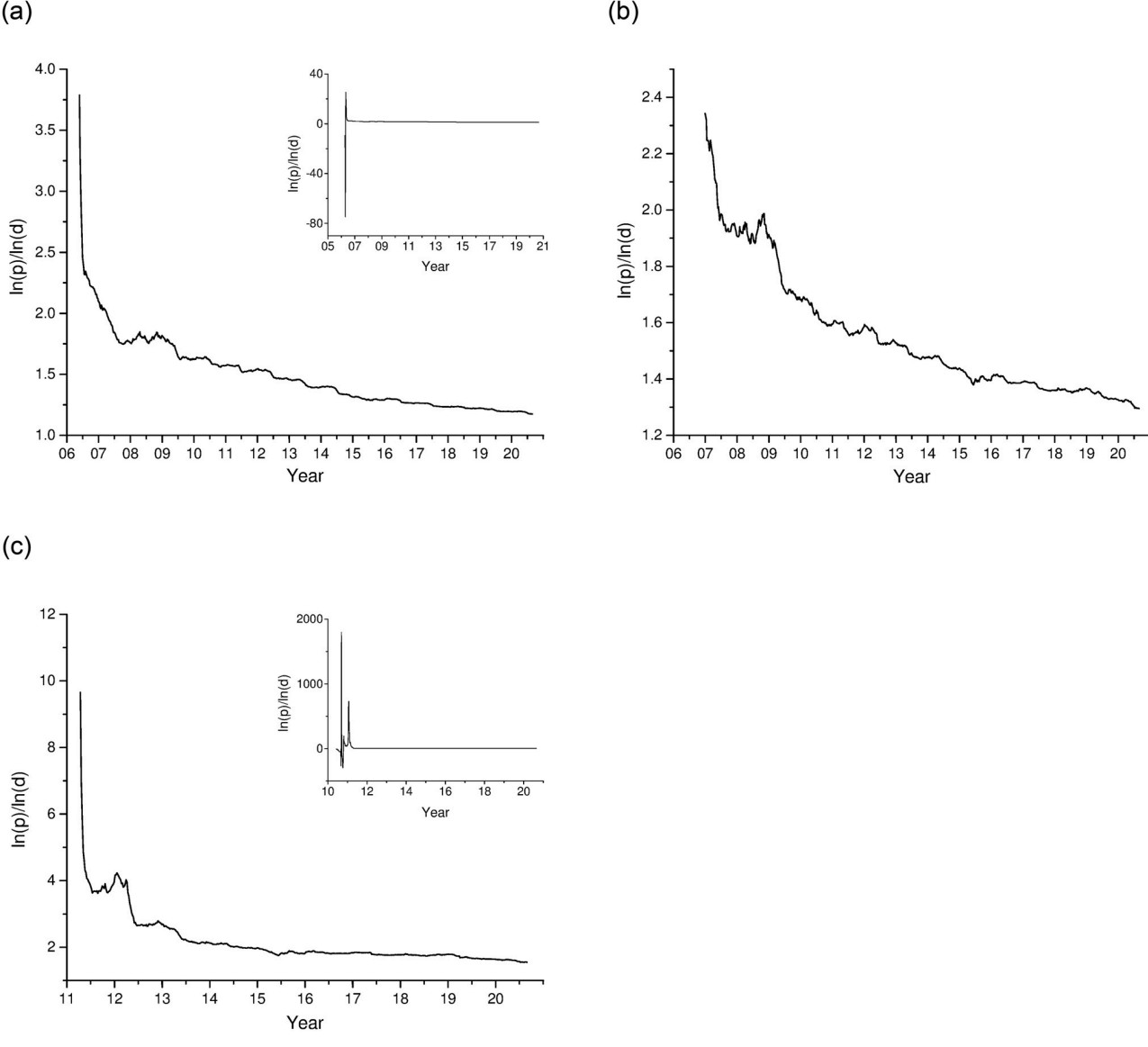

**Fig 4. The ratios of the real stock prices' natural logarithm to the real dividends' natural logarithm.** This figure plots the $\ln(p)/\ln(d)$ series for three sectors: (a) CSI 300 from May 26, 2006 to August 28, 2020, (b) SMEs board from December 29, 2006 to August 28, 2020, (c) GEM from April 15, 2011 to August 28, 2020. All series data are weekly. And entire samples of CSI 300 and GEM are shown in the small maps at the top right of each. Source: Wind database, and website of the National Bureau of Statistics of the People's Republic of China.

series are calculated. Then, the lag order $k$ in Eq (8) is set to be 0, and the trend term is not included in this model. Under this setting, the accumulative sum series is performed for 2000 times Monte Carlo simulations corresponding to each test with $r_0 = 0.01+1.8/T^{1/2}$, so as to determine critical value series at a significance level of 0.05.

## 4.2 Bubble existence test

In this section, SADF and GSADF are used to test $\ln(p)/\ln(d)$ series to verify the existence of bubbles in China's multi-level stock market. The detailed results are shown in Table 1. The SADF test statistics of CSI 300, SMEs and GEM are -25.7813, -2.8319 and -4.8915 respectively. They do not exceed any of their critical values, so there is no evidence that the $\ln(p)/\ln(d)$ series

**Table 1. Results for SADF and GSADF test on ln($p$)/ln($d$) of CSI 300, SMEs and GEM.**

| Sectors | Test statistics | | Critical Values | | |
|---|---|---|---|---|---|
| | | | 90% | 95% | 99% |
| CSI 300 | SADF | -25.7813 | 1.2510 | 1.5256 | 2.0173 |
| | GSADF | 3.7641 | 2.0530 | 2.2873 | 2.7947 |
| SMEs | SADF | -2.8319 | 1.2301 | 1.5603 | 2.0030 |
| | GSADF | 3.3651 | 2.0115 | 2.2582 | 2.8018 |
| GEM | SADF | -4.8915 | 1.1765 | 1.4585 | 1.9523 |
| | GSADF | 3.1144 | 1.9926 | 2.2171 | 2.8321 |

Critical values are all implemented by 2000 Monte Carlo simulations. These tests are conducted by MATLAB software.

have explosive behavior by SADF test. The GSADF test statistics of CSI 300, SMEs and GEM are 3.7641, 3.3651, and 3.1144 respectively, which are larger than their respective 99% critical values. Sufficient evidence shows that the ln($p$)/ln($d$) series of three sectors are explosive, indicating the existence of bubbles in China's stock market.

## 4.3 PWY time-stamping

PWY time-stamping strategy is to date bubble points by comparing right-tail recursive ADF statistic series with their simulated 95% ADF critical value series. Stamping results are shown in Fig 5. It is clear that PWY time-stamping strategy based on SADF test does not identify any bubbles in CSI 300, SME and GEM markets. It is obvious that the ability of this method to identify multiple bubbles is limited.

## 4.4 BSADF time-stamping

The right-tail backward recursive BSADF statistic sequences of three sectors are compared with their corresponding 95% BSADF critical value sequences, so as to date the periodical bubbles of China's stock market. The dating results are shown in Fig 6.

As described in section 2.6, we restrict that the duration of a bubble must exceed 14 weeks. The short-term fluctuations lasting less than 14 weeks cannot be called bubble. The real price series are also plotted in Fig 6, which are only used as a reference for the bubble curves. As shown in Fig 6, with almost every bubble boom, peak and burst, real price series rise to a peak within a small range and then fall. This reflects the fact that changes in real price series can also convey part of information about bubble process and can serve as reference series.

Further details of each bubble are shown in Table 2. It can be seen that China's Main-Board Market has identified 8 bubbles since 2006, namely 09W25-09W39, 12W25-12W47, 13W23-14W01, 14W27-15W35, 16W29-16W50, 17W26-18W06, 19W26-19W46, and 20W27-unclosed, and the degree and duration of each bubble are different. Bubble (1) 09W25-09W39

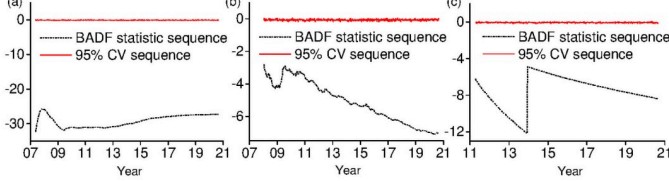

**Fig 5. PWY time-stamping results.** This figure plots the BADF statistic series and their simulated 95% critical value series for three sectors: (a) CSI 300, (b) SMEs, and (c) GEM.

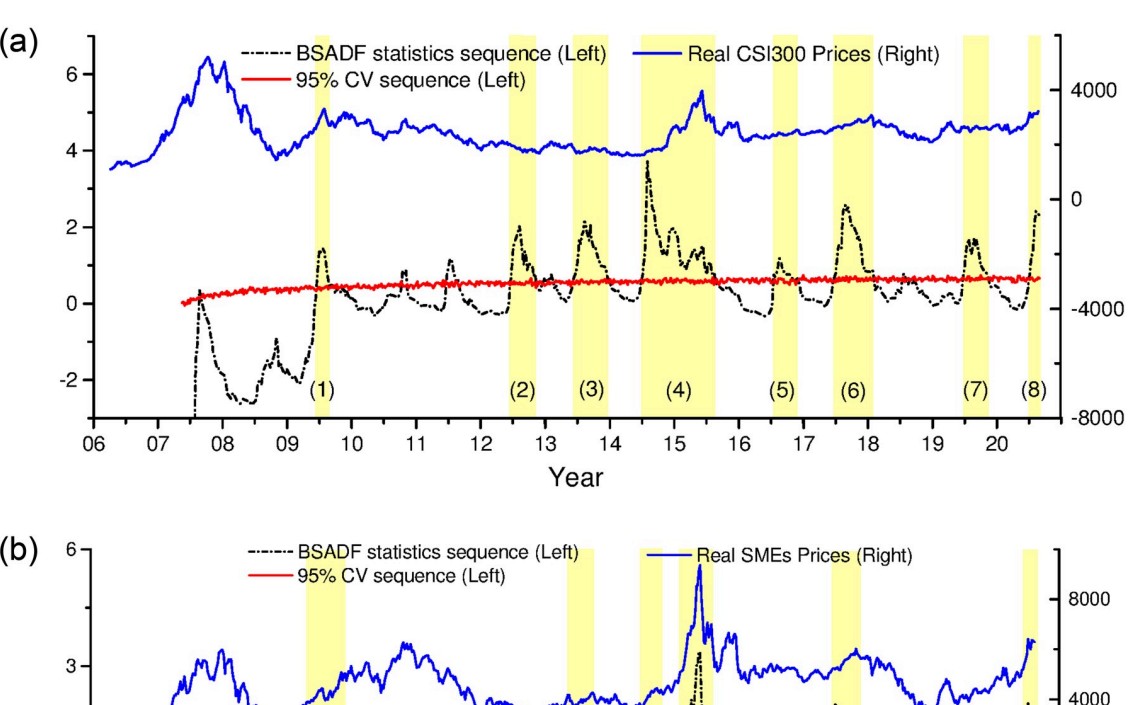

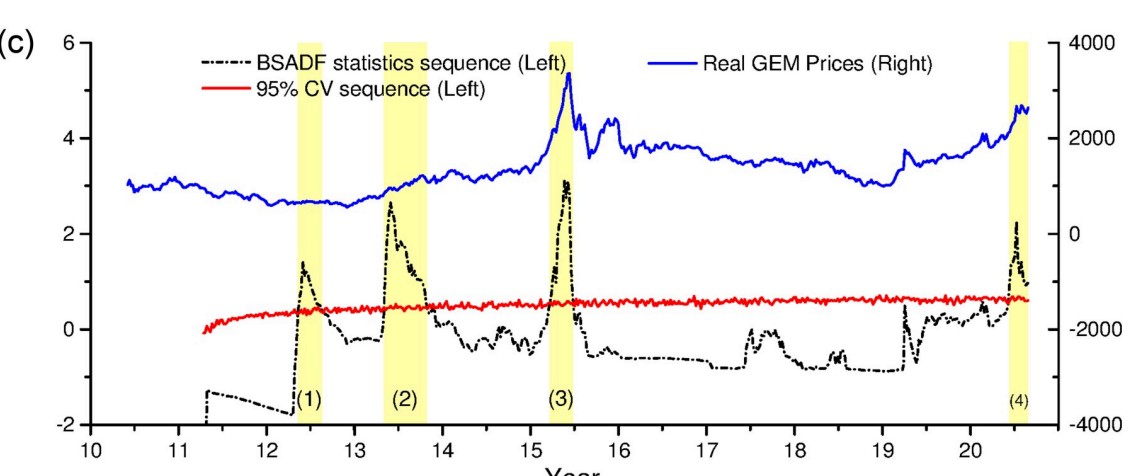

**Fig 6. BSADF time-stamping results.** This figure plots the BSADF statistic series (left scale), simulated 95% critical value series (left scale), and real price series (right scale) for three sectors: (a) CSI 300, (b) SMEs, and (c) GEM.

**Table 2. The detailed bubble dating results of CSI 300, SMEs and GEM.**

| Sectors | No. & Bubble period | Origination | Peak(date) | Termination | Duration |
|---|---|---|---|---|---|
| CSI 300 | (1) 09W25-09W39 | Jun 19,2009 | 1.4522(Jul 24,2009) | Sep 25,2009 | 15 weeks |
| | (2) 12W25-12W47 | Jun 15,2012 | 2.0323(Aug10,2012) | Nov16,2012 | 22 weeks |
| | (3) 13W23-14W01 | Jun 07,2013 | 2.1967(Aug09,2013) | Jan 03,2014 | 31 weeks |
| | (4) 14W27-15W35 | Jul 04,2014 | 3.7641(Aug01,2014) | Aug28,2015 | 61 weeks |
| | (5) 16W29-16W50 | Jul 15,2016 | 1.1820(Aug19,2016) | Dec09,2016 | 21 weeks |
| | (6) 17W26-18W06 | Jun 23,2017 | 2.5672(Aug25,2017) | Feb09,2018 | 33 weeks |
| | (7) 19W26-19W46 | Jun 28,2019 | 1.6947(Jul 26,2019) | Nov15,2019 | 21 weeks |
| | (8) 20W27-unclosed | Jul 03,2020 | 2.4146(Aug07,2020) | Unclosed | Unclosed |
| SMEs | (1) 09W19-09W51 | May 8,2009 | 1.3199(Jun19,2009) | Dec18,2009 | 33weeks |
| | (2) 13W22-13W43 | May31,2013 | 1.0444(Jul 12,2013) | Oct 25,2013 | 22 weeks |
| | (3) 14W28-14W46 | Jul 11,2014 | 1.6006(Aug22,2014) | Nov14,2014 | 19 weeks |
| | (4) 15W07-15W34 | Feb 13,2015 | 3.3651(Jun12,2015) | Aug21,2015 | 28 weeks |
| | (5) 17W27-17W50 | Jun 30,2017 | 2.0129(Jul 14,2017) | Dec08,2017 | 23 weeks |
| | (6) 20W23- unclosed | Jun 05,2020 | 2.0842(Jul 10,2020) | Unclosed | Unclosed |
| GEM | (1) 12W20-12W36 | May11,2012 | 1.4123(Jun01,2012) | Aug31,2012 | 17 weeks |
| | (2) 13W19-13W44 | May10,2013 | 2.6613(May31,2013) | Nov01,2013 | 26 weeks |
| | (3) 15W13-15W27 | Mar 27,2015 | 3.1144(May22,2015) | Jul 03,2015 | 15 weeks |
| | (4) 20W13- unclosed | Jun 12,2020 | 2.2654(Jul 10,2020) | Unclosed | Unclosed |

lasts for 15 weeks and is the shortest. Bubble (5) 16W29-16W50 is the smallest with a peak of 1.1820. The (4) 14W27-15W35 bubble is the largest and longest, with a peak of 3.7641 and duration of 61 weeks. The (6) 17W26-18W06 bubble is the second largest, lasting 33 weeks with a peak of 2.5672.

Six bubbles are identified in SMEs board, namely 09W19-09W51, 13W22-13W43, 14W28-14W46, 15W07-15W34, 17W27-17W50, and 20W23- unclosed. Among them, the bubble in 2015 is the largest, with a peak of 3.3651, and lasting for 28 weeks. The 2009 bubble lasts for the longest, reaching 33 weeks.

China's GEM has been established for a relatively short time. Since its establishment in 2009, there have been four bubbles, namely, 12W20-12W36, 13W19-13W44, 15W13-15W27, and 20W13- unclosed. The largest one is the bubble in 2015, which peaks at 3.1144 and lasts from March 27, 2015 to July 3, 2015. The longest bubble appeared in 2013, which lasts for 26 weeks.

## 4.5 Results analysis

Any bull market is liquidity-driven and presupposes a loose credit cycle, with which there is at least one structural move. From Table 2, it is clear that there is a bubble in 2009 in both CSI 300 and SMEs. The occurrence of this bubble was related to the "4 trillion Yuan" economic stimulus plan, credit incentive policy and industrial revitalization policy successively launched at the end of 2008. These policies led to a sharp rebound in Renminbi credit and flooding of liquidity, resulting in a bubble in the stock market.

From 2012 to 2013, many non-standardized projects can obtain double-digit risk-free returns. From the perspective of the growth rate of social finance, it can also be defined as a round of loose credit cycle, which led to the creation of bubbles in three sectors. Furthermore, 2012–2013 is a structural bull market for GEM.

There is a great bull market that comes fast and goes fast from 2014 to the first half of 2015. The exception is that this is a credit contraction cycle. At the end of 2013, because of the

decline of real estate, the restrain of local infrastructure and the suppressant of non-standardized debt assets, wealth management products and residential departments began to look for alternatives to non-standardized debt assets, leading to the increase of leverage in stock market. Large scale leveraged funds brought by margin financing and securities lending entered the stock market, which promoted the surge of investor sentiment. In addition, bubbles of this period occurred during the transition period of China's economy. At this time, old economy driven by exports and investment has been growing weak, but new economic growth point has not yet been formed. In order to achieve a smooth economic transformation, the central bank continued to implement loose monetary policy, and a large amount of capital poured into financial market, reducing the risk-free interest rate and also stimulating the improvement of investors' risk preference. There are two main reasons for the burst of bubbles in this period: firstly, the lack of supervision over shareholders' off-market capital allocation behavior by regulatory authorities in early stage and the rush and rough "deleveraging" in late stage caused the funding to flee, leading to the break of capital chain. Secondly, malicious short selling and excessive issuance of new shares have severely hit market confidence and caused investors' expectations to burst, leading to the collapse of bubbles.

From 2016 to 2017, although the supply-side reform caused an excess capacity reduction on small and medium-sized enterprises, the housing price rise in first-tier cities and the shanty town reform in third-tier and fourth-tier cities led to a loose credit cycle. Therefore, bubbles appeared in the Main-Board Market and SMEs board.

It is worth mentioning that there are bubbles in all three sectors in mid-2020, and those bubbles have not yet ended as of August 28, 2020. There must be a reason why China's stock market is still soaring this year in the troubled situation. Under the impact of Corona Virus Disease 2019 (COVID-19), China implemented a loose monetary policy. The resumption of work and production is slow, especially for small and medium-sized enterprises, but credit easing has directed support to small, medium and micro enterprises, causing dislocation of credit cycle and economic cycle. As a result, fluidity siltation is formed, which will most likely flow into financial market. Neither the bond market nor trust products have performed as expected this year, so more money poured into the more profitable stock market. Furthermore, among the major countries in the world, China has the best control of the epidemic and the fastest economic recovery. Therefore, as a basic market with good recovery momentum and low-valued depressions in the world, peripheral loose liquidity has poured into China A shares.

There are respectively 8 bubbles, 6 bubbles and 4 bubbles in CSI 300, SMEs and GEM markets. Why does China's stock market experience multiple bubbles in a short period of time? Comparing the dating results of this paper with the results of PSY [18] on S&P 500 series, it can be found that after more than 100 years of development, US stock market has evolved from an emerging market with frequent fluctuations in initial period to a relatively stable mature capital market determined by fundamentals, and investors' investment behavior is more rational. But China's stock market is still in the stage of emerging market due to its short establishment time. In contrast, China's market fluctuates frequently, not only with multiple bubbles, but also with short period. In China's capital market, many investors have speculative psychology, and most of them choose short-term operations to profit from the price difference. They are sensitive to market reaction and follow the trend seriously [39], which indicates a more serious "herd effect". In general, there are several reasons for the frequent occurrence of bubbles:

**(1) The capital market is emerging.**    At present, China's multi-level capital market has been established for a short time, the imperfection of market system and the lack of standardization leads to the trading mechanism far from effective market level. Secondly, the lack of effective delisting system makes investors more likely to have irrational speculation.

Furthermore, the monopoly or blockade occurring in investors' acquisition of information caused by incomplete information and information asymmetry will increase the blindness of investment. Therefore, China's capital market should actively improve the construction of market economy, improve the timeliness of information disclosure, and guide small and medium-sized investors to make rational investments.

**(2) Herd behavior among investors.** Individual investors account for the majority of China stock market. Compared with institutional investors, small and medium-sized individual investors lack professional knowledge and investment skills, and have a serious herd mentality. They pay more attention to the investment behavior of others and blindly follow suit. The "herd effect", an irrational behavior of learning and imitation, will lead to positive feedback transactions in the stock market investment [40]. It will easily aggravate the sharp fluctuations of market, and then lead to the formation of short-term bubbles [41]. Therefore, it is necessary to strengthen the education of investors. In addition, institutional investors have strong asset scale and technological advantages; they can optimize the return and risk of investment in the market through professional teams. So, it is also necessary to continuously increase the number of institutional investors.

**(3) The bubble warning mechanism is insufficient.** Frequent emergence of bubbles reflects the shortage of timely bubble risk monitoring mechanism. Therefore, it is difficult to introduce bubble control measures in time. So, bubble monitoring and early warning mechanism is very necessary. By using BSADF time stamping strategy to detect the periodical bubbles in real time, the occurrence and collapse of bubbles can be warned at the first time, so as to provide timely and accurate experience support for bubble control and adjustment of relevant policies.

**(4) The impact of government regulatory system.** Because of the speculative behavior of Chinese investors, government has set up a variety of restrictions on stock market to ensure normal operation for the sake of avoiding investors' excessive risk-taking. Such as: the price limit system, the time when listed companies can be traded or not traded, and the entry criteria for overseas investors. On the one hand, the government implicitly guarantees the stock market; on the other hand, it strictly regulates the market. This will not only weaken investors' risk awareness, but also distort the operation mechanism and relationship between supply and demand of market. To a large extent, this has led to a sharp shock in stock market, exacerbating the formation of short-term bubbles. Therefore, it is necessary to constantly improve the government's regulatory mechanism and establish an effective delisting mechanism. Only by establishing a market system suitable for long-term investment and value investment can the existing speculative market be transformed into an investment market.

## 5. Conclusions

In application of GSADF bubble presence test and BSADF time stamping, the start and end points of recursive subsamples are variable, which increases the number and range of subsamples, and thus can effectively date the points of periodical bubbles in real time. Based on the financial present value model, this paper applies right-tail recursive ADF test to test multiple bubbles in China's multi-level stock market. Unlike the other researches in China, the ratios of the real stock prices' natural logarithm to the real dividends' natural logarithm are used for our testing instead of stock price index. From perspective of combination of stock price and dividend, multiple bubbles are detected and dated in real time. The following conclusions are drawn.

Firstly, the results of GSADF test fully indicate that there are bubbles in China's stock market. The results of BSADF time-stamping show that there are 8 bubbles in China's Main-Board Market, 6 bubbles in SMEs board, and 4 bubbles in GEM, respectively.

Secondly, these bubbles are liquidity-driven and presuppose a loose credit cycle, with the exception of the bubble in 2014–2015, which is a credit contraction cycle. Bubbles in 2009 is related to the "4 trillion Yuan" economic stimulus plan, and 2012–2013 is a structural bull market for GEM. While the largest bubble in 2014–2015 is driven entirely by leverage. From 2016 to 2017, the housing price rise in first-tier cities and the shanty town reform in third-tier and fourth-tier cities lead to a loose credit cycle. Bubbles in 2020 were formed from the fluidity siltation caused by dislocation of credit cycle and economic cycle during the epidemic.

In addition, the emergence of multiple bubbles in a short period of time indicates that China's stock market is still emerging capital market with frequent fluctuations. And "herd effect" will aggravate sharp fluctuations. This also reflects the lack of monitoring mechanism for bubble risk.

Finally, based on above analysis, the following suggestions are proposed for preventing stock market bubbles: improving the timeliness of information disclosure, strengthening education for investors, establishing bubble monitoring and early warning mechanism, and constructing a market system suitable for long-term investment and value investment.

## Supporting information

**S1 File. Data of simulated stationary time series, random walk process and explosive autoregressive process.**
(XLSX)

**S2 File. CPI Deflator of three sectors.**
(XLSX)

**S3 File. Raw data for testing.**
(XLSX)

**S4 File. Data of testing results.**
(XLSX)

## Author Contributions

**Conceptualization:** Ge Li, Ming Xiao.

**Data curation:** Ge Li.

**Methodology:** Ge Li.

**Software:** Ge Li, Xionghui Yang.

**Supervision:** Ming Xiao.

**Validation:** Xionghui Yang, Ying Guo.

**Writing – original draft:** Ge Li.

**Writing – review & editing:** Ming Xiao, Shengyi Yang.

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
