## [Decision Letter · Decision Letter 0]

7 Jan 2021

PONE-D-20-36507

Research on multiple bubbles in China's multi-level stock market

PLOS ONE

Dear Dr. Xiao,

Thank you for submitting your manuscript to PLOS ONE. After careful consideration, we have decided that your manuscript does not meet our criteria for publication and must therefore be rejected.

I am sorry that we cannot be more positive on this occasion, but hope that you appreciate the reasons for this decision.

Yours sincerely,

Haijun Yang, Ph.D

Academic Editor

PLOS ONE

Additional Editor Comments (if provided):

I regret to inform you that the reviewers recommend against publishing your manuscript, and I must therefore reject it.

Reviewers' comments:

Reviewer's Responses to Questions

**Comments to the Author**

1. Is the manuscript technically sound, and do the data support the conclusions?

Reviewer #1: No

Reviewer #2: Yes

2. Has the statistical analysis been performed appropriately and rigorously? 

Reviewer #1: No

Reviewer #2: Yes

3. Have the authors made all data underlying the findings in their manuscript fully available?

Reviewer #1: No

Reviewer #2: Yes

4. Is the manuscript presented in an intelligible fashion and written in standard English?

Reviewer #1: No

Reviewer #2: Yes

5. Review Comments to the Author

Reviewer #1: This paper applies right-tail recursive ADF test to test multiple bubbles in China's multi-level stock market. After a careful reading of this manuscript, I would recommend to reject this paper.

Key comments:

1. The topic is not interesting.

2. Critical values should be generated to account for the presence of non-stationary volatility. Hence, all the analysis in this paper is flawed.

3. This paper has numerous grammar and language issues.

Reviewer #2: Am very grateful for inviting me to review this manuscript. The Paper Title: "Research on multiple bubbles in China's multi-level stock market" is well written and constructed. The Author have conducted a thorough literature review. Have analyze information accurately by doing Empirical test.

I therefore recommend that the paper should be accepted.

6. PLOS authors have the option to publish the peer review history of their article (what does this mean?). If published, this will include your full peer review and any attached files.

Reviewer #1: No

Reviewer #2: No

- - - - -

---

## [Author Response · Author response to Decision Letter 0]

28 Jan 2021

Dear Dr. Yang:

Thank you for your efficient work in processing our manuscript entitled “Research on multiple bubbles in China's multi-level stock market” (Manuscript No: PONE-D-20-36507). After we have carefully read the comments from the reviewers, we realize that the major merits of our work were not fully identified or recognized by the reviewers. Here we would respond to the reviewers’ comments (the replies are highlighted in blue).

Reviewer #1:

1.The topic is not interesting.

Response：As China becomes the second largest economy in the world, research on China's capital market has become increasingly important. Besides, as a representative emerging market, research on China's capital market can help people understand the characteristics of emerging markets. China's stock market is characterized by a dominance of inexperienced individual investors, binding short‐sales constraints (lifted only in 2011), a small asset float (before the split‐share reform of 2005–2006) and heavy share turnover despite high transaction costs. Individual investors in China's stock market are less informed and more subject to behavioral biases than institutional investors. All these factors make very likely the presence of active speculative behavior in Chinese stock market. Therefore, our work is meaningful.

Most researches in China only adopt price series to detect capital market bubble without considering dividends. It does not conform to the specification of test model and cannot be theoretically supported. For example, Company A's stock price is 100 Yuan per share, cash dividend is 10 Yuan per share, while company B's stock price is 10 Yuan per share, cash dividend is 1 cent per share. At this time, there are more likely bubbles in B's share price. If only considering the price 100>10, it is biased to say there is a bubble in company A's stock price. Different from them, we select the ratios of the natural logarithm of real stock prices to the natural logarithm of real dividends to conduct tests from perspective of combination of stock price and dividend. Therefore, our work is different and interesting.

2. Critical values should be generated to account for the presence of non-stationary volatility. Hence, all the analysis in this paper is flawed.

Response：Recursive explosive‐root (vs random walk) tests (Phillips, Shi, & Yu, 2015a,b) with a unit root (market fundamental) null and an explosive (bubble) alternative enable researchers to detect the rise and collapse of bubbles on a given asset market. It is shown by Phillips, Shi, and Yu (2015a) that the PSY method outperforms the other recursive methods. The PSY procedure aims to detect the local explosive dynamics of speculative bubbles. The principle and reliability of this method are described in detail in articles “EXPLOSIVE BEHAVIOR IN THE 1990s NASDAQ: WHEN DID EXUBERANCE ESCALATE ASSET VALUES?*” (Phillips, Wu, & Yu, 2011) and “TESTING FOR MULTIPLE BUBBLES: HISTORICAL EPISODES OF EXUBERANCE AND COLLAPSE IN THE S&P 500∗” (Phillips, Shi, & Yu, 2015a). The statistical analysis in our work are appropriate and rigorous.

3. This paper has numerous grammar and language issues.

Response：We have found English professionals to read our article and made a lot of grammatical changes. All the changes have been marked red in the revised manuscript. The modifications are as follows.

(1) Page 2, line 24-26:

ORIGINAL: Understanding bubble phenomenon and dating the origination and termination points in real time can provide an early warning diagnosis for financial bubbles and help regulatory authorities to control it and maintain market order.

REVISED: Understanding bubble phenomenon and dating the period of bubbles in real time can provide an early warning diagnosis for financial bubbles and help regulatory authorities to control it and maintain market order.

(2) Page 3, line 55-57:

ORIGINAL: Rational bubble theory assumes that the actual price of an asset is equal to the present value of relevant fundamentals and the bubble component that is expected to grow at the real interest rate, and that investors have rational expectations.

REVISED: Rational bubble theory assumes that the actual price of an asset is equal to the sum of the present value of relevant fundamentals and the bubble component which is expected to grow at the real interest rate, meanwhile the investors have rational expectations.

(3) Page 3, line 57-59:

ORIGINAL: Early rational bubble test based on the assume that bubble was linear, namely, if there was a bubble, it would always exist, and would not burst or start again.

REVISED: Early rational bubble test based on the assume that the bubble was linear, which meant that the bubble would always exist and would not burst or start again.

(4) Page 4, line 69-70:

ORIGINAL: In addition, if there is a relatively short-term bubble in a long time series of data, it may not correctly identify the presence of bubble.

REVISED: In addition, if there is a relatively short-term bubble in a long time series of data, the existence of this bubble may not be correctly recognized.

(5) Page 4, line 71:

ORIGINAL: The above methods have imposed very little structure on bubbles. Actually, bubbles have certain theoretical properties can be clearly used for detection.

REVISED: However, there are certain theoretical properties that can be clearly used for bubbles detection.

(6) Page 4, line 74-77:

ORIGINAL: Diba and Grossman [12-13] observed that rational bubble could not start (or restart); therefore, if it exists now, it must exist on the first day of trading and always exist. If the bubble "pops", it must collapse to zero because of the lack of arbitrage opportunities and the impossibility of negative prices.

REVISED: Diba and Grossman [12-13] observed that rational bubble could not start (or restart); therefore, if there is a bubble, it must exist on the first day of trading and always exist. If the bubble "pops", the bubble must collapse to zero due to the lack of arbitrage opportunities and the impossibility of negative prices.

(7) Page 5, line 91-92:

ORIGINAL: But the Markov model is computationally expensive. And, in some cases, the asymptotic distribution of unit root test statistic calculated by this model is impossible to analyze.

REVISED: However, the Markov model is computationally expensive, and the asymptotic distribution of unit root test statistic calculated by this model is impossible to analyze in some cases.

(8) Page 5, line 93-95:

ORIGINAL: In recent years, the breakthrough of periodical bubble test is that they apply right-tailed DF tests of unit root null hypothesis against explosive autoregression alternative hypothesis to relevant series in levels only.

REVISED: In recent years, the breakthrough of the periodical bubble test is that the right-tailed DF tests of unit root null hypothesis against explosive autoregression alternative hypothesis is applied to relevant series in levels only.

(9) Page 7, line 134-138:

ORIGINAL: The rest of this paper is organized as follows. Section 2 discusses the basic model specification of bubble test and the principles of right-tail recursive ADF test, including SADF, GSADF and BSADF. Section 3 introduces the sample processing, and provides related preliminary data analysis. The empirical bubble test results and result analysis are presented in Section 4. Section 5 concludes this paper.

REVISED: The rest of this paper is organized as follows. In Section 2, the basic model specification of bubble test and the principle of right-tail recursive ADF test are discussed, including SADF, GSADF and BSADF. In Section 3, the sample processing is introduced and the relevant preliminary data analysis is provided. The empirical bubble test results and result analysis are introduced in Section 4. In Section 5, the full text is summarized and conclusions are drawn.

(10) Page 10, line 201-202:

ORIGINAL: So the null hypothesis and the alternative hypothesis are respectively:

REVISED: Therefore, the null hypothesis and the alternative hypothesis are:

(11) Page 25, line 477-478:

ORIGINAL: By contrast, China's market fluctuates frequently, not only does it have multiple bubbles, but each time the bubble starts and bursts rapidly.

REVISED: In contrast, the China's market fluctuates frequently, not only with multiple bubbles, but also with short period.

(12) Page 25, line 492-495:

ORIGINAL: Compared with institutional investors, small and medium-sized individual investors lack professional knowledge and investment skills and have a serious herd mentality. They are more concerned about other people's investment behavior and blindly follow the trend.

REVISED: Compared with institutional investors, small and medium individual investors lack professional knowledge and investment skills, and have a serious herd mentality. They pay more attention to the investment behavior of others and blindly follow suit.

4. Have the authors made all data underlying the findings in their manuscript fully available?

Reviewer #1: No

Response：We made all data underlying the findings in our manuscript fully available. We provided the data in supporting information of our manuscript.

We understand that the misunderstanding might be caused by the unclear description in our manuscript, but we believe that the results are of merit and the paper is potentially publishable in the journal. Therefore, we would be most grateful if you could re-consider our work and give us a second opportunity. Thank you very much for your patience and understanding. If you have any question about this paper, please don’t hesitate to let me know. Looking forward to hearing from you.

Kind regards,

Ge Li, Ph.D candidate

Corresponding author: Ming Xiao, Ph.D

E-mail address: blue_ridge111@yahoo.com

---

## [Decision Letter · Decision Letter 1]

12 May 2021

PONE-D-20-36507R1

Research on multiple bubbles in China's multi-level stock market

PLOS ONE

Dear Dr. Xiao,

Thank you for submitting your manuscript to PLOS ONE. After careful consideration, I feel that it has merit but does not fully meet PLOS ONE’s publication criteria as it currently stands. 

When I accepted this appeal manuscript I carefully read the comments of the reviewer and the decision of my colleague. In my case, I was not able to take a decision just with the previous report considering that the reject was not argued. All decisions have to be motivated by reviewers as well as by editors.

I have invited two new reviewers and now, based on their comments, I consider that this manuscript is suitable of publication if some minor concerns are addressed.

In my opinion this paper is suitable of publication in Plos One and I would like to apologize to the authors for the inconvenient caused. Therefore, I invite you to submit a revised version of the manuscript that addresses the points raised during the review process.

We look forward to receiving your revised manuscript.

Kind regards,

J E. Trinidad Segovia

Academic Editor

PLOS ONE

Journal Requirements:

When submitting your revision, we need you to address these additional requirements.Please ensure that your manuscript meets PLOS ONE's style requirements, including those for file naming. The PLOS ONE style templates can be found at

Reviewers' comments:

Reviewer's Responses to Questions

**Comments to the Author**

1. If the authors have adequately addressed your comments raised in a previous round of review and you feel that this manuscript is now acceptable for publication, you may indicate that here to bypass the “Comments to the Author” section, enter your conflict of interest statement in the “Confidential to Editor” section, and submit your "Accept" recommendation.

Reviewer #3: (No Response)

Reviewer #4: All comments have been addressed

2. Is the manuscript technically sound, and do the data support the conclusions?

Reviewer #3: Yes

Reviewer #4: Yes

3. Has the statistical analysis been performed appropriately and rigorously? 

Reviewer #3: Yes

Reviewer #4: Yes

4. Have the authors made all data underlying the findings in their manuscript fully available?

Reviewer #3: Yes

Reviewer #4: Yes

5. Is the manuscript presented in an intelligible fashion and written in standard English?

Reviewer #3: No

Reviewer #4: Yes

6. Review Comments to the Author

Reviewer #3: The paper titled "Research on multiple bubbles in China's multi-level stock market", authored by Ge Li, Ming Xiao, Xionghui Yang, Ying Guo and Shengyi Yang, is an interesting work on detecting the existence of bubbles in China's stock market. As indicated by the authors, this paper applies right-tail recursive ADF test to test multiple bubbles in China's multi-level stock market. Unlike other researchers, the ratio of the natural logarithm of real stock prices to the natural logarithm of real dividends has been selected, instead of stock price indices.

However, I have some concerns before recommending possible acceptation of this manuscript:

1. The paper contains numerous grammatical mistakes which should be immediately fixed (the list is not exhaustive):

* Line 26: Change "test" to "tests".

* Line 29: Change "Different from" to "Unlike".

* Line 32: in the main board?

* Line 34: credit easing cycle?

* Line 45: Change "bubble" to "bubbles".

* Line 50: inflexibility? Please, define or clarify.

* Line 58: Change "assume" to "assumption".

* Line 65: he explicitly put the bubble in alternative hypotheses? Please, explain.

* Etcetera.

2. There are several theoretical reasoning which should be more explained:

* Lines 73-74: "If stock prices and dividends [...] in asset price".

* Lines 77-78: "Through Dickey-Fuller [...] no bubbles".

* Lines 87-89: This sentence does not make sense.

* Lines 102-103: This sentence does not make sense.

3. The statements included in the Subsection "Model specification" should be more supported by theoretical literature because this paper introduces the novelty of the ratio ln(p)/ln(d) and this advices to relate this indicator to other parameters involved in the analysis of the necessary and sufficient conditions for the existence of bubbles. To do this, the authors may consult the following references:

* Sethi, S.P.; Derzko, N.A.; Lehoczky, J.P. Mathematical analysis of the Miller-Modigliani theory.

Operations Research Lett. 1982, 1 (4), 148–152.

* Sethi, S.P.; Derzko, N.A.; Lehoczky, J.P. A stochastic extension of the Miller-Modigliani

framework. Mathematical Finance 1991, 1 (4), 57–76.

*Sethi, S.P.; Derzko, N.A.; Lehoczky, J.P. When does the share price equal the present value of

future dividends? Economic Theory 1996, 8, 307–319.

* Cruz Rambaud, S. (2013). Arbitrage Theory with State-Price Deflators, Stochastic Models, 29:3, 306-327.

Reviewer #4: In this paper, the authors are focused on exploring financial bubbles in China's multi-level stock market. In this regard, they consider the ratio between the (log) prices of real stocks and the (log) of real dividends, instead of stock price indices to carry out their calculations, unlike other papers which also investigated bubbles in China's stock market. As a result, they state that they are a number of bubbles in China's main board (8), Growth Enterprise Market (4), and Small and Medium Enterprises Board (6). Such conclusions were obtained on the basis of the right-tail recursive ADF test.

May the authors should take a look at the following reference:

\\bibitem{}

M.~Fern{\\'{a}}ndez-Mart{\\'{i}}nez, M.A. S{\\'{a}}nchez-Granero, M.J.

{Mu{\\~{n}}oz Torrecillas}, and B.~McKelvey, \\emph{{A COMPARISON OF THREE

HURST EXPONENT APPROACHES TO PREDICT NASCENT BUBBLES IN S{\\&}P500 STOCKS}},

Fractals \\textbf{25} (2017), no.~1,

where the transition from efficient market behavior (EMB, hereafter) to the beginning of a bubble in the S&P500 index was explored. With this aim, another technique, based on the self-similarity exponent, was applied. In fact, they found that the higher (resp., the lower) the self-similarity index of the series, the higher (resp., the lower) the mean of the price changes, and hence, the better (resp., the worse) the performance of that stock. As such, the beginning of the transition from EMB to herding behavior could be identified.

In summary, I found the content of this manuscript technically sound with its conclusions being supported by the analyses and calculations that were carried out. Thus, I recommend it for acceptance after taking into account my previous comment.

7. PLOS authors have the option to publish the peer review history of their article (what does this mean?). If published, this will include your full peer review and any attached files.

Reviewer #3: No

Reviewer #4: **Yes: **M. Fernández-Martínez

---

## [Author Response · Author response to Decision Letter 1]

26 Jun 2021

To academic editor and reviewers,

PLOS ONE

Dear Prof. J E. Trinidad Segovia and Reviewers:

Thank you for giving us a chance to resubmit the paper and your constructive comments on our manuscript entitled "Research on multiple bubbles in China's multi-level stock market" (ID: PONE-D-20-36507R1). 

Those comments are very helpful for revising and improving our paper. We have studied the comments carefully and made corrections which we hope meet with approval. Efforts were also made to correct the grammatical mistakes in the manuscript. We mark all the changes in red in the revised manuscript with track changes. Attached please find our revised manuscript. Listed below are our point-by-point responses to the editor and reviewers’ comments (the replies are highlighted in blue).

Responds to academic editor:

Comment 1: Please review your reference list to ensure that it is complete and correct. If you have cited papers that have been retracted, please include the rationale for doing so in the manuscript text, or remove these references and replace them with relevant current references. Any changes to the reference list should be mentioned in the rebuttal letter that accompanies your revised manuscript. If you need to cite a retracted article, indicate the article’s retracted status in the References list and also include a citation and full reference for the retraction notice.

Response: We have reviewed our reference list; all the reference are complete and correct. We didn’t cite papers that have been retracted. We made some changes in our references:

we deleted reference [16]: Van Norden S. Regime switching as a test for exchange rate bubbles. J. Appl. Econom. 1996; 11(3): 219-251. doi: 10.1002/(SICI)1099-1255(199605)11:3<219::AID-JAE394>3.0.CO;2-S.

we added references:

[33]. Sethi SP. When does the share price equal the present value of future dividends? Econ. Theory. 1996; 8(2): 307-319. doi: 10.1007/BF01211820.

[34]. Sethi SP, Derzko NA, Lehoczky JP. A stochastic extension of the Miller-Modigliani framework. Math. Finance. 1991; 1(4): 57-76. doi: 10.1111/j.1467-9965.1991.tb00019.x.

[35]. Sethi SP, Derzko NA, Lehoczky JP. Mathematical analysis of the Miller-Modigliani theory. Operations Res. Lett. 1982; 1(4): 148-152. doi: 10.1016/0167-6377(82)90018-9.

[37]. Rambaud SC. Arbitrage theory with state-price deflators. Stoch. Models. 2013; 29(3): 306-327. doi: 10.1080/15326349.2013.808902.

[41]. Fernández-Martínez M, Sánchez-Granero MA, Muñoz Torrecillas MJ, Mckelvey B. A comparison of three Hurst exponent approaches to predict nascent bubbles in S&P500 stocks. Fractals. 2017; 25(01): 1750006-105. doi: 10.1142/S0218348X17500062.

The detail changes of our references are elaborated in the following responds to the reviewers.

Comment 2: When submitting your revision, we need you to address these additional requirements. Please ensure that your manuscript meets PLOS ONE's style requirements, including those for file naming.

Response: After reading the PLOS ONE style templates carefully, we ensure that our manuscript meets PLOS ONE's style requirements, including those for file naming.

Modification 1: We have added a second affiliation to co-author Xionghui Yang. The detailed information of this affiliation is added in page 1, line 5 of the revised manuscript: CITIC Group Corporation, Beijing, Beijing, People's Republic of China.

Modification 1: We have renumbered the equation after equation (10). The number of equation in page 14, line 279 is changed from (12) to (11); and the equation in the next line is changed from (13) to (12). The number of equation in page 15, line 295 is changed from (14) to (13). We are very sorry for our carelessness.

Responds to the reviewer’s comments:

Reviewer #3:

Comment 1: The paper contains numerous grammatical mistakes which should be immediately fixed (the list is not exhaustive):

* Line 26: Change "test" to "tests".

Response: According to the reviewer’s comment, we have changed "test" to "tests" in line 26. Furthermore, we have fixed the similar mistakes in the paper.

* Line 29: Change "Different from" to "Unlike".

Response: Thank you for your valuable advice. We have changed "Different from" to "Unlike" in line 29.

* Line 32: in the main board?

Response: We have changed “the main board” to “the Main-Board Market” in line 32. It refers to the traditional stock market, and is the main place for the issuance, listing and trading of securities in China.

* Line 34: credit easing cycle?

Response: We have changed “credit easing cycle” to “loose credit cycle”, which is the opposite of the tight credit cycle.

* Line 45: Change "bubble" to "bubbles".

Response: We have changed "bubble" to "bubbles" in line 45. We have also fixed the similar mistakes in the other part of the paper.

* Line 50: inflexibility? Please, define or clarify.

Response: In page 3 line 50, we changed “all showed the inflexibility of bubbles” to “are all the recognized bubble in the world”. The original statement is not accurate.

* Line 58: Change "assume" to "assumption".

Response: Thank you for your careful work. We have changed " assume " to " assumption " in line 58.

* Line 65: he explicitly put the bubble in alternative hypotheses? Please, explain.

Response: We have changed “he explicitly put the bubble in alternative hypotheses” to “the existence of bubbles was explicitly considered as alternative hypotheses for the first time.”. Detailed explanation: the alternative hypothesis of West's two-step tests was hypothesis that there was a bubble, in this test, the bubble was first explicitly considered as alternative hypotheses.

We are sorry for grammatical mistakes in this paper. We have carefully checked the manuscript vocabulary and grammar again. There are also some other revises in the revision:

(1) on Page 2, line 30, " we select the ratios of the natural logarithm of real stock prices to the natural logarithm of real dividends instead of stock price indices to conduct tests. " has been revised as " the ratios of the real stock prices' natural logarithm to the real dividends' natural logarithm are used for our testing instead of stock price index. "

(2) On Page 2, line 37, "can" has been revised as "not only". 

(3) On Page 2, line 38, "and" has been revised as "but". 

(4) On Page 3, line 61, "and" has been revised as ",". 

(5) On Page 3, line 63, " Marsh and Merton [8] suggested that variance bound tests would fail when dividends and stock prices were non-stationary. At the same time, there were small sample distortions. " has been revised as " Marsh and Merton [8] suggested that variance bound tests would fail when dividends and stock prices were non-stationary, and along with the small sample distortions. ". 

(6) On Page 4, line 68, “rejecting null hypotheses too frequently” has been revised as " rejecting null hypotheses frequently ".

(7) On Page 5, line 93, “impossible” has been revised as " hard ".

(8) On Page 5, line 94, “In recent years, the breakthrough of the periodical bubble test is that the right-tailed DF tests of unit root null hypothesis against explosive autoregression alternative hypothesis is applied to relevant series in levels only.” has been revised as " In recent years, the breakthrough of the periodical bubbles test is that the right-tailed DF test is applied to relevant series. The null hypothesis is unit root process, and the alternative hypothesis is explosive autoregression process. ". 

(9) On Page 5, line 101, “can” has been revised as " could ".

(10) On Page 6, line 106, “In addition, the estimates are consistent” has been revised as " , and the estimated points are consistent with the actual points ".

(11) On Page 6, line 118, “One simple way” has been revised as " A simpler method ".

(12) On Page 7, line 131, “ratios of the logarithm of real stock prices to the logarithm of real dividends rather than stock price indices are selected as target sequences,” has been revised as " in this paper, ratios of the real stock prices' natural logarithm to the real dividends' natural logarithm are selected as target sequences ".

(13) On Page 7, line 132, “times” has been added.

(14) On Page 10, line 199, “traditional ADF tests will confuse test results.” has been revised as " the results of traditional ADF test will be confusing".

(15) On Page11, line 217, “with” has been revised as "and".

(16) On Page11, line 227, “Further” has been revised as " Furthermore ".

(17) On Page12, line 244, “its test ability will be reduced” has been revised as " the applicability of this method will be reduced ".

(18) On Page13, line 256, “between BSADF and GSADF” has been added.

(19) On Page16, line 334, “this paper selects the ratios of the natural logarithm of real stock prices to the natural logarithm of real dividends” has been revised as "the ratios of the real stock prices' natural logarithm to the real dividends' natural logarithm (i.e. ln(p)/ln(d)) are selected ".

(20) On Page17, line 337, “dividends are less or even not, resulting in large fluctuations of the initial ln(p)/ln(d)” has been revised as " dividends are less, or even no dividends, which results in large fluctuations of the initial ln(p)/ln(d)".

(21) On Page17, line 344, Fig 4 Title has been revised from “Fig 4. The ratios of the natural logarithm of real stock prices to the natural logarithm of real dividends.” to " Fig 4. The ratios of the real stock prices' natural logarithm to the real dividends' natural logarithm.".

(22) On Page 18, line 358, “trend” has been added.

(23) On Page 18, line 369, “respectively” has been added.

(24) On Page 18, line 373, “times” has been added.

(25) On Page 18, line 378, “times” has been added.

(26) On Page 19, line 385, “it cannot be said” has been revised as " there is no evidence that ".

(27) On Page 20, line 395, “We can see” has been revised as " It is clear ".

(28) On Page 20, line 396, “this method has limited ability to identify multiple bubbles” has been revised as " the ability of this method to identify multiple bubbles is limited ".

(29) On Page 20, line 407, “In BSADF dating Results Fig 6, real price series are also plotted” has been revised as " The real price series are also plotted in Fig 6 ".

(30) On Page 22, line 427, “The longest is the bubble of 2013” has been revised as " The longest bubble appeared in 2013 ".

(31) On Page 22, line 430, “we can see” has been revised as " it is clear that ".

(32) On Page 24, line 458, “So, we can see bubbles in main board” has been revised as " Therefore, bubbles appeared in the Main-Board Market ".

(33) On Page 24, line 460, “there is a bubble in the middle of 2020 in three sectors, and as of August 28, 2020, it is not over” has been revised as " there are bubbles in all three sectors in mid-2020, and those bubbles have not yet ended as of August 28, 2020 ".

(34) On Page 24, line 461, “Why China's stock market soared this year despite many disasters? There must be a reason for this.” has been revised as " There must be a reason why China's stock market is still soaring this year in the troubled situation ".

(35) On Page 24, line 463, “The resumption of work and production by entities, especially small and medium-sized enterprises, is slow” has been revised as " The resumption of work and production is slow, especially for small and medium-sized enterprises ".

(36) On Page 27, line 528, “Different from the other researchers in China, we select the ratios of the natural logarithm of real stock prices to the natural logarithm of real dividends instead of stock price indices to conduct tests” has been revised as " Unlike the other researches in China, the ratios of the real stock prices' natural logarithm to the real dividends' natural logarithm are used for our testing instead of stock price index ".

(37) On Page 27, line 532, “First, GSADF test results” has been revised as " Firstly, the results of GSADF test ".

(38) On Page 27, line 535, “Second” has been revised as " Secondly ".

(39) On Page 28, line 540, “Bubbles in 2020 thanks to” has been revised as " Bubbles in 2020 were formed from ".

(40) On Page 28, line 542, “Third” has been revised as " In addition ".

(41) On Page 28, line 545, “Fourth” has been revised as "Finally ".

Comment 2: There are several theoretical reasoning which should be more explained:

* Lines 73-74: "If stock prices and dividends [...] in asset price".

Response: We have changed “If stock prices and dividends are cointegrated, they share the same stochastic drift, and there is no bubble in asset price.” to “If first differences of stock prices have a stationary mean and/or stock prices are cointegrated with dividends, there would be evidence against the existence of rational bubbles.”

The theoretical reasoning is explained as follow (the details of theoretical model are in section 2.1 of our paper):

Of particular interest to researchers in this area are “rational bubbles”. The concept of rational bubble can be illustrated by financial present value theory, where the real price of an asset is assumed to be equal to the present value of relevant fundamentals and a bubble component that grows in expectation at the real interest rate, and investors are assumed to have rational expectations. Under these assumptions investing in the asset can be a rational choice for investors even though its current observed price is higher than the price level that is justified by relevant fundamentals.

Diba and Grossman (1988) implement stationarity tests for explosive rational bubbles in stock prices using a model that assumes a constant discount rate, but that allows unobservable variables to affect market fundamentals and also allows different valuations of expected capital gains and expected dividends. If the first differences of the unobservable variables and the first differences of dividends are stationary (in the mean) and first differences of stock prices are stationary, then the model implies that rational bubbles do not exist. If the levels of the unobservable variables and the first differences of dividends are stationary, and stock prices and dividends are cointegrated of order (1,1), then rational bubbles do not exist.

Diba and Grossman point out evidence that first differences of stock prices have a stationary mean and/or evidence that stock prices are cointegrated with dividends would be evidence against the existence of rational bubbles. Please refer to article “Explosive Rational Bubbles in Stock Prices?” (Diba and Grossman, 1988).

* Lines 77-78: "Through Dickey-Fuller [...] no bubbles".

Response: In page 4, line 78, We have changed “Through Dickey-Fuller test, they found that both dividends and stock prices were stationary in differences, indicating no bubbles.” to “They applied the Dickey-Fuller (DF) test to the price and dividend series in levels and first differences, then found that both stock prices and dividends were stationary in first differences. Since differencing an explosive autoregressive process does not lead to a stationary process, a rejection from the DF test for the first difference of the price and dividend series, suggests that no rational bubble exists.”

The detail theoretical reasoning of this point is the same as the last point, so we won't repeat it here.

* Lines 87-89: This sentence does not make sense.

Response: According to the reviewer’s comment, we have deleted the sentence“The initial popular method is the two-regime Markov-switching unit root test [16], which considers the expanding and collapsing periods of bubbles as two different regimes and allows the bubble to switch between two regimes.”. With this operation, we delete the corresponding reference [16]: Van Norden S. Regime switching as a test for exchange rate bubbles. J. Appl. Econom. 1996; 11(3): 219-251. doi: 10.1002/(SICI)1099-1255(199605)11:3<219::AID-JAE394>3.0.CO;2-S.

* Lines 102-103: This sentence does not make sense.

Response: Considering the Reviewer’s suggestion, we have changed the sentence “on the basis of PWY, they used right recursive unit root test with flexible window width, namely Generalized Sup ADF, to test the existence of bubbles” to “namely Generalized Sup ADF with flexible window width”.

Comment 3: The statements included in the Subsection "Model specification" should be more supported by theoretical literature because this paper introduces the novelty of the ratio ln(p)/ln(d) and this advices to relate this indicator to other parameters involved in the analysis of the necessary and sufficient conditions for the existence of bubbles. To do this, the authors may consult the following references:

Response: Thank you very much for your advice on Model specification. The references that you recommend discuss an explicit necessary and sufficient condition on the dividend stream of a publicly traded company, under which the price of the company's share is equal to the present value of the future dividends that will accrue to it. These researches are very valuable. So, we have cited these references in our paper. The details are as below:

* Sethi, S.P.; Derzko, N.A.; Lehoczky, J.P. Mathematical analysis of the Miller-Modigliani theory. Operations Research Lett. 1982, 1 (4), 148–152.

Response: We have cited this reference in page 7 line 143 in our paper, and this reference is listed in the reference list as number [35].

* Sethi, S.P.; Derzko, N.A.; Lehoczky, J.P. A stochastic extension of the Miller-Modigliani framework. Mathematical Finance 1991, 1 (4), 57–76.

Response: We have cited this reference in page 7 line 143 in our paper, and this reference is listed in the reference list as number [34].

*Sethi, S.P.; Derzko, N.A.; Lehoczky, J.P. When does the share price equal the present value of future dividends? Economic Theory 1996, 8, 307–319.

Response: We have cited this reference in page 7 line 143 in our paper, and this reference is listed in the reference list as number [33].

* Cruz Rambaud, S. (2013). Arbitrage Theory with State-Price Deflators, Stochastic Models, 29:3, 306-327.

Response: This reference developed some mathematical results for the existence of asset price bubbles, which is consistent with the martingale analysis of financial markets. More specifically, this article studies the relation between the divergence of the sum of dividend-price ratios and the absence of bubbles. So, we have cited this reference in page 9 line 175 in our paper, and this reference is listed in the reference list as number [37].

Reviewer #4:

Comment 1: May the authors should take a look at the following reference:

\\bibitem{}M.~Fern{\\'{a}}ndez-Mart{\\'{i}}nez, M.A. S{\\'{a}}nchez-Granero, M.J.{Mu{\\~{n}}oz Torrecillas}, and B.~McKelvey, \\emph{{A COMPARISON OF THREE HURST EXPONENT APPROACHES TO PREDICT NASCENT BUBBLES IN S{\\&}P500 STOCKS}},Fractals \\textbf{25} (2017), no.~1,

where the transition from efficient market behavior (EMB, hereafter) to the beginning of a bubble in the S&P500 index was explored. With this aim, another technique, based on the self-similarity exponent, was applied. In fact, they found that the higher (resp., the lower) the self-similarity index of the series, the higher (resp., the lower) the mean of the price changes, and hence, the better (resp., the worse) the performance of that stock. As such, the beginning of the transition from EMB to herding behavior could be identified.

Response: The reference you recommend explored whether there are some connections between the self-similarity exponent of a stock (as a Herding Behavior indicator) and the stock’s future performance under the assumption that the HB will last for some time. The findings of this reference are significant. We have cited this reference in page 25 line 497 in our paper, and this reference is listed in the reference list as number [41].

We tried our best to improve the manuscript according to the reviewer’s suggestion. Again, we appreciate for the Editors and Reviewer’s time and help for improving the quality of our manuscript. We hope that the correction will meet with approval. If you have any question about this paper, please don’t hesitate to let me know. 

Looking forward to hearing from you.

Kind regards,

Ge Li

Corresponding author: Ming Xiao

E-mail address: blue_ridge111@yahoo.com

---

## [Decision Letter · Decision Letter 2]

19 Jul 2021

Research on multiple bubbles in China's multi-level stock market

PONE-D-20-36507R2

Dear Dr. Xiao,

We’re pleased to inform you that your manuscript has been judged scientifically suitable for publication and will be formally accepted for publication once it meets all outstanding technical requirements.

Kind regards,

J E. Trinidad Segovia

Section Editor

PLOS ONE

Additional Editor Comments (optional):

Reviewers' comments:

Reviewer's Responses to Questions

**Comments to the Author**

1. If the authors have adequately addressed your comments raised in a previous round of review and you feel that this manuscript is now acceptable for publication, you may indicate that here to bypass the “Comments to the Author” section, enter your conflict of interest statement in the “Confidential to Editor” section, and submit your "Accept" recommendation.

Reviewer #3: All comments have been addressed

2. Is the manuscript technically sound, and do the data support the conclusions?

Reviewer #3: Yes

3. Has the statistical analysis been performed appropriately and rigorously? 

Reviewer #3: Yes

4. Have the authors made all data underlying the findings in their manuscript fully available?

Reviewer #3: Yes

5. Is the manuscript presented in an intelligible fashion and written in standard English?

Reviewer #3: Yes

6. Review Comments to the Author

Reviewer #3: The authors have adequately addressed all my former comments and suggestions. In my opinion, the paper has gained more quality and accuracy.

7. PLOS authors have the option to publish the peer review history of their article (what does this mean?). If published, this will include your full peer review and any attached files.

Reviewer #3: No

---

## [Editor Report · Acceptance letter]

23 Jul 2021

PONE-D-20-36507R2 

Research on multiple bubbles in China's multi-level stock market 

Dear Dr. Xiao:

I'm pleased to inform you that your manuscript has been deemed suitable for publication in PLOS ONE. Congratulations! Your manuscript is now with our production department. 

Kind regards, 

on behalf of

Dr. J E. Trinidad Segovia 

Section Editor

PLOS ONE